# A Gradient-based Kernel Approach for Efficient Network Architecture Search

## Abstract

It is widely accepted that vanishing and exploding gradient values are the main reason behind the difficulty of deep network training. In this work, we take a further step to understand the optimization of deep networks and find that both gradient correlations and gradient values have strong impacts on model training. Inspired by our new finding, we explore a simple yet effective network architecture search (NAS) approach that leverages gradient correlation and gradient values to find well-performing architectures. To be specific, we first formulate these two terms into a unified gradient-based kernel and then select architectures with the largest kernels at initialization as the final networks. The new approach replaces the expensive "train-then-test" evaluation paradigm with a new lightweight function according to the gradient-based kernel at initialization. Experiments show that our approach achieves competitive results with orders of magnitude faster than "train-then-test" paradigms on image classification tasks. Furthermore, the extremely low search cost enables its wide applications. It also obtains performance improvements on two text classification tasks.[1]

## 1 Introduction

Understanding and improving the optimization of deep networks has been an active field of artificial intelligence. One of the mysteries in deep learning is why extremely deep neural networks are hard to train. Currently, the vanishing and exploding gradient problem is widely believed to be the main answer (Bengio et al., 1994; Hochreiter & Schmidhuber, 1997; Pascanu et al., 2013). Gradients exponentially decrease (or increase) from the top layer to the bottom layer in a multi-layer network. By extension, the current vanishing and exploding gradient problem also refers to extremely small (or large) gradient values. Following this explanation, several widely-used training techniques have been proposed to assist optimization. These studies can be roughly classified into four categories: initialization-based approaches (Glorot & Bengio, 2010; He et al., 2015; Zhang et al., 2019), activation-based approaches (Hendrycks & Gimpel, 2016; Klambauer et al., 2017), normalization based approaches (Ioffe & Szegedy, 2015; Salimans & Kingma, 2016; Ulyanov et al., 2016; Lei Ba et al., 2016; Nguyen & Salazar, 2019), and skip-connection-based approaches (He et al., 2016; Huang et al., 2017a). These techniques provide good alternatives to stabilize gradients and bring promising performance improvements.

Despite much evidence showing the importance of steady gradient values, we are still curious about whether there are unexplored but important factors. To answer this question, we conduct extensive experiments and find some surprising phenomena. First, the accuracy curves of some models converge well with the remaining gradient vanishing and exploding problem. Second, we find several cases where models with similar gradient values show different convergence performance. These results indicate that gradient values may not be vital as much as we expect, and some hidden factors play a significant role in optimization. To answer this question, we explore other gradient features, such as covariance matrix, correlations, variance, and so on. These features are used to evaluate gradients from different perspectives. The covariance matrix evaluates the similarity between any two parameters, which is widely used in optimization studies (Zhao & Zhang, 2015; Faghri et al., 2020). Gradient correlation is defined to evaluate the similarity between any two examples. Among them, experiments show that gradient correlation is an important factor. Compared to widely-explored

---

[1]We will release the code on publication.

gradient values, gradient correlations are rarely studied. As gradient values affect the step size in optimization, vanishing gradients prevent models from changing weight values. In the worst case, this could completely stop training. Gradient correlations evaluate the randomness of gradient directions in Euclidean space. Lower gradient correlations indicate more "random" gradient directions and more "conflicting" parameter updates. For a better understanding, we visualize absolute gradient values and gradient correlations (See Section 4 for more details). It illustrates that models with either small values or small correlations show worse convergence and generalization performance, indicating the importance of these two factors.

Following the new finding, we take a further step to explore gradient correlations and gradient values in network architecture search. We first formulate two factors into a unified gradient-based kernel, which is defined as the average of a gradient dot-product matrix. Motivated by the observation that architectures with larger kernels at initialization tend to have better average results, we develop a lightweight network architecture search approach, called GT-NAS, which evaluates architectures according to the gradient-based kernel at initialization. Unlike other NAS approaches that evaluate architectures via full training, which takes massive computation resources, GT-NAS only relies on a few examples for kernel calculation. We first select top-$k$ architectures with largest kernels as candidates, which are then trained to select the best one based on validation performance. In practice, $k$ is usually set to be a very small value. Experiments show that GT-NAS achieves competitive results with extremely fast search on NAS-Bench-201 (Ying et al., 2019), a NAS benchmark dataset. The low search cost allows us to apply NAS on diverse tasks. Specifically, the structure searched by the proposed policy outperforms the naive baseline without NAS on 4 datasets covering image classification and text classification.

The main contributions are summarized as follows:

- We propose that both gradient correlations and gradient values matter in optimization, which gives a new insight to understand and develop optimization techniques.
- Following the new finding, we propose a gradient-based kernel approach for NAS, which is able to search for well-performing architectures efficiently.
- Experiments show that GT-NAS achieves competitive results with orders of magnitude faster than the naive baseline.

## 2 RELATED WORK

**Understanding and improving optimization** Understanding and improving the optimization of deep networks has long been a hot research topic. Bengio et al. (1994) find the vanishing and exploding gradient problem in neural network training. To address this problem, a lot of promising approaches have been proposed in recent years, which can be classified into four research lines.

The first research line focuses on initialization (Sutskever et al., 2013; Mishkin & Matas, 2016; Hanin & Rolnick, 2018). Glorot & Bengio (2010) propose to control the variance of parameters via appropriate initialization. Following this work, several widely-used initialization approaches have been proposed, including Kaiming initialization (He et al., 2015), Xaiver initialization (Glorot & Bengio, 2010), and Fixup initialization (Zhang et al., 2019). The second research line focuses on normalization (Ioffe & Szegedy, 2015; Lei Ba et al., 2016; Ulyanov et al., 2016; Wu & He, 2018; Nguyen & Salazar, 2019). These approaches aim to avoid the vanishing gradient by controlling the distribution of intermediate layers. The third is mainly based on activation functions to make the derivatives of activation less saturated to avoid the vanishing problem, including GELU activation (Hendrycks & Gimpel, 2016), SELU activation (Klambauer et al., 2017), and so on. The motivation behind these approaches is to avoid unsteady derivatives of the activation with respect to the inputs. The fourth focuses on gradient clipping (Pascanu et al., 2013).

**Gradient-based Kernel** Our work is also related to gradient-based kernels (Advani & Saxe, 2017; Jacot et al., 2018). NTK (Jacot et al., 2018) is a popular gradient-based kernel, defined as the Gram matrix of gradients. It is proposed to analyze model's convergence and generalization. Following these studies, many researchers are devoted to understand current networks from the perspective of NTK (Lee et al., 2019; Hastie et al., 2019; Allen-Zhu et al., 2019; Arora et al., 2019). Du et al. (2019b) use the Gram matrix of gradients to prove that for an $m$ hidden node shallow neural network

with ReLU activation, as long as $m$ is large enough, randomly initialized gradient descent converges to a globally optimal solution at a linear convergence rate for the quadratic loss function. Following this work, Du et al. (2019a) further expand this finding and prove that gradient descent achieves zero training loss in polynomial time for a deep over-parameterized neural network with residual connections.

**Network architecture search** Neural architecture search aims to replace expert-designed networks with learned architectures (Chang et al., 2019; Li et al., 2019; Zhou et al., 2020; He et al., 2020a; Fang et al., 2020; He et al., 2020b; You et al., 2020; Alves & de Oliveira, 2020). The first NAS research line mainly focuses on random search. The key idea is to randomly evaluate various architectures and select the best one based on their validation performance. However, despite promising results, many of them require thousands of GPU days to achieve desired results. To address this problem, Ying et al. (2019) propose a reinforcement learning based search policy which introduces an architecture generator with validation accuracy as reward. Another research line is based on evolution approaches. Real et al. (2019) propose a two-stage search policy. The first stage selects several well-performing parent architectures. The second stage applies mutation on these parent architectures to select the best one. Following this work, So et al. (2019) apply the evolution search on Transformer networks and achieve new state-of-the-art results on machine translation and language modeling tasks. Although these approaches can reduce exploration costs, the dependence on validation accuracy still leads to huge computation costs. To get rid of the dependence on validation accuracy, several studies (Jiang et al., 2019; Liu et al., 2019; Dong & Yang, 2019; Zela et al., 2020; Chu et al., 2020) re-formulate the task in a differentiable manner and allow efficient search using gradient descent. Unlike these studies, we propose a lightweight NAS approach, which largely reduces evaluation cost. Furthermore, the new approach does not rely on additional search models, thus also saves memory usage.

## 3 NOTATIONS: GRADIENT VALUE AND GRADIENT CORRELATION

Assume $\boldsymbol{g}(l)_i = \{g(l)_{i,1}, \cdots, g(l)_{i,r}\}$ is the gradient vector of example $i$ with respect to all parameters in the $l$-th layer. $r$ is the total number of parameters. We combine all gradient vectors together to construct a gradient matrix $\boldsymbol{G}(l) \in \mathcal{R}^{d \times r}$ where $d$ refers to the number of training examples. We define a matrix $\boldsymbol{P}(l)$ to store the dot-product of gradients. The dot-product between two gradient vectors is:

$$\boldsymbol{P}(l)_{i,j} = \langle \boldsymbol{G}(l)_i, \boldsymbol{G}(l)_j \rangle. \tag{1}$$

We divide $\boldsymbol{P}(l)$ by the norm of gradient vectors to get a gradient correlation matrix $\boldsymbol{C}(l)$:

$$\boldsymbol{C}(l)_{i,j} = \frac{\boldsymbol{P}(l)_{i,j}}{||\boldsymbol{G}(l)_i|| \cdot ||\boldsymbol{G}(l)_j||}. \tag{2}$$

Then, we use the average value of $\boldsymbol{C}(l)$ as the final correlation score:

$$R = \mathbb{E}(\boldsymbol{C}(l)). \tag{3}$$

Here we also evaluate the absolute gradient value:

$$V = \mathbb{E}(|\boldsymbol{G}(l)|). \tag{4}$$

Since it requires too much time to calculate $R$ and $V$ for plenty of training examples, we randomly sample $\mu$ examples to estimate $R$ and $V$ in implementation for fast calculation. The dot-product between vectors with too many elements will cause the overflow problem. We also randomly sample $\theta$ parameters when calculating $R$. To avoid the sampling bias that might affect our findings, we test different $\mu \in [50, 100, 200, 300, 400, 500]$ and $\theta \in [50, 100, 200, 300, 400, 500]$, and observe the same phenomena. For simplicity, we show our findings in the next section with $\mu = 50$ and $\theta = 50$.

## 4 RETHINKING VANISHING AND EXPLODING GRADIENTS

In this section, we first analyze how the problem of vanishing and exploding gradient arises. Then, we show some experimental cases to prove that vanishing and exploding gradient values are not the only factors behind the difficulty of deep network training.

### 4.1 VANISHING AND EXPLODING GRADIENTS IN TRAINING DEEP NETWORKS

We consider a simple multi-layer fully-connected neural network with $L$ layers. Assume that the output of $h$-th layer is:

$$\boldsymbol{y}^{(h)} = \sigma(\boldsymbol{w}^{(h)}\boldsymbol{y}^{(h-1)}), \tag{5}$$

where $\sigma$ is the activation function and $\boldsymbol{w}^{(h)}$ is the weight matrix. The gradient for the $h$-th layer with respect to the weight matrix $\boldsymbol{w}^{(h)}$ is:

$$\frac{\partial \mathcal{L}}{\partial \boldsymbol{w}^{(h)}} = \langle \frac{\partial \mathcal{L}}{\partial \boldsymbol{y}^{(L)}}, \boldsymbol{y}^{(h-1)} \frac{\partial \boldsymbol{y}^{(L)}}{\partial \boldsymbol{y}^{(h)}} \boldsymbol{J}^{(h)} \rangle = \langle \frac{\partial \mathcal{L}}{\partial \boldsymbol{y}^{(L)}}, \boldsymbol{y}^{(h-1)} (\prod_{k=h+1}^{L} \boldsymbol{J}^{(k)}\boldsymbol{w}^{(k)})\boldsymbol{J}^{(h)} \rangle, \tag{6}$$

where $\mathcal{L}$ is the loss function and $\frac{\partial \mathcal{L}}{\partial \boldsymbol{y}^{(L)}}$ is the derivative of the output of the last layer[2]. $\boldsymbol{y}^{(h-1)}$ is the output of $h-1$ layer. $\boldsymbol{J}^{(k)}$ is the diagonal matrix where the main diagonal is the list of the derivatives of the activation with respect to the inputs in the $k$-th layer:

$$\boldsymbol{J}^{(k)} = \mathbf{diag}(\sigma'(\boldsymbol{w}_1^{(k)}\boldsymbol{y}^{(k-1)}), \cdots, \sigma'(\boldsymbol{w}_d^{(k)}\boldsymbol{y}^{(k-1)})) \in \mathcal{R}^{d \times d}, \tag{7}$$

where $d$ is the dimension of the activation and $\boldsymbol{w}_d^{(k)}$ is the $d$-th row in matrix $\boldsymbol{w}^{(k)}$ in the $k$ layer.

From this equation, we can see that the norm of weight matrix $\boldsymbol{w}$, the output of previous layer $\boldsymbol{y}^{(h-1)}$ and the derivative of activation play an important role in controlling the gradient stability. Most previous studies mainly focus on these key terms to address the vanishing and exploding problem.

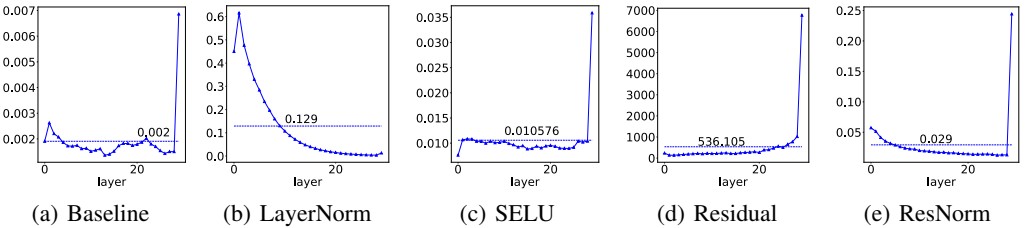

|          |          |          |          |          |
|----------|----------|----------|----------|----------|
| (a) Baseline | (b) LayerNorm | (c) SELU | (d) Residual | (e) ResNorm |

Figure 1: Gradient values on different models. The X-axis is the index of layers and Y-axis denotes the average absolute gradient value in each layer. The dotted line shows the average of all absolute gradients in a model. "Baseline" has extremely low gradient values on all layers and its gradients are multiplied by 1000 for clear presentation. Even with advanced training techniques, the vanishing and exploding problem is not addressed completely. The gradients between the top layer and the bottom layer are still in orders of magnitude in "LayerNorm", "Residual", and "ResNorm".

### 4.2 GRADIENT VALUES AND GRADIENT CORRELATIONS BOTH MATTER

To empirically evaluate the effects of unsteady gradient values, we implement widely-used optimization techniques, including activation-based techniques (e.g., SELU), normalization-based techniques (e.g., LayerNorm), and skip-connection-based techniques (e.g., Residual). We adopt a 32-layer multi-layer perceptron (MLP) network as the backbone for MNIST (LeCun et al., 2010), an image classification dataset. The bottom is a 2-layer convolutional component and the top is a 30-layer feed-forward component. The baseline model uses ReLU activation and Xavier initialization (Glorot & Bengio, 2010). The evaluated techniques are then added into the baseline. All the models use the same hyper-parameter settings for a fair comparison. We use the AdaGrad optimizer (Duchi et al., 2011) with a learning rate $1.0$ and adopt a learning rate decay mechanism with $\gamma = 0.7$. The batch size is set to $64$. All experiments run on a single NVIDIA V100 GPU. The total number of running epochs is $15$.

Figure 1 shows gradient value curves in different layers at initialization, and Figure 2 demonstrates the distribution over gradient values at initialization and at training stages. "Baseline" has serious gradient vanishing problems with extremely small values in all layers at initialization. With the

---

[2]Here we adopt numerator layout notation.

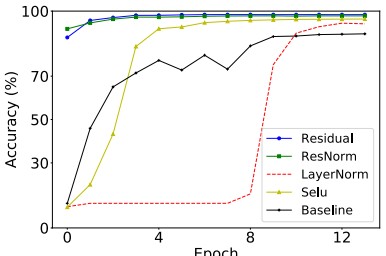

Figure 2: From left to right, we demonstrate the charts of gradient values at initialization and at training stages, gradient correlations at initialization, and at training stages. "Baseline" and "Layer-Norm" with either lower gradient correlations or lower gradient values achieve worse convergence performance. "ResNorm" and "Residual" with higher gradient correlations or higher gradient values, obtain better convergence performance.

increase of training steps, the vanishing problem is slightly addressed but still serious. It is as expected that "Baseline" achieves the worst convergence performance with $89.52$ test accuracy, as shown in Figure 3. With advanced training optimization techniques, some layers have larger gradient values. The accuracy curves of "SELU", "Residual" and "ResNorm" converge well. These results verify the importance of steady gradient values in optimization. However, there are still some cases that are hard to understand from the perspective of gradient values. First, it is out of our expectation that the accuracy curves of "LayerNorm", "Residual", and "ResNorm" converge well although the gap of gradients between the top layer and the bottom layer in these models is still in orders of magnitude. These results demonstrate that a deep network is robust to vanishing or exploding gradients to some extent. Second, "SELU" and "ResNorm" with similar ranges of gradient values show distinct convergence speed. It indicates that there are still unknown factors behind the optimization of deep networks.

In addition to gradient values, we observe that gradient correlations contribute to the convergence performance. We compute gradient correlations based on Eq. 2. Figure 2 shows gradient correlation distributions. Compared to "SELU" and "ResNorm", "LayerNorm" has much lower gradient correlations and shows worse convergence performance. Despite extremely low gradient values, "Baseline"with higher average gradient correlations at initialization achieves faster convergence than "LayerNorm". These results demonstrate that gradient correlations also matter in optimization.

Figure 3: Test accuracy of different models on CIFAR10.

# 5 LEVERAGING GRADIENT CORRELATIONS AND GRADIENT VALUES IN NETWORK ARCHITECTURE SEARCH

We also explore how to leverage this new finding to improve the performance of deep networks in real-world tasks. First, we unify these two items into a gradient-based kernel, defined as the average of the Gram matrix of gradient, i.e., $\boldsymbol{P}$ in Eq.1. Starting with several simple architectures, one observation is that networks with larger kernels at initialization tend to achieve higher average convergence performance. Motivated by this observation, we propose to adopt the gradient-based kernel on NAS to search for the optimal architecture.

First, we generate $s$ possible architectures as the search space $\mathcal{S}$. In this work, we do not fix the generating policy to keep the flexibility on diverse tasks and scenarios. For each architecture, we calculate the gradient-based kernel based on Eq.1 and select $k$ architectures with the highest scores as candidates, which are then trained from scratch to get their results on a validation set. For fast calculation, we set $\mu = 50$ and $\theta = 50$ when estimating kernel. The architecture with the highest validation accuracy is selected as the final architecture. We normalize gradients by subtracting the average of sampled gradients before calculation. The new evaluation function does not need any training steps. Compared to other NAS approaches that train hundreds of architectures, the new approach largely reduces search costs.

## 6 EXPERIMENTS

In this section, we evaluate the proposed approach on NAS-Bench-201, a NAS benchmark dataset. It provides a fair setting for evaluating different search techniques.

### 6.1 DATASETS

NAS-Bench-201 is a benchmark dataset for NAS algorithms, constructed on image classification tasks, including CIFAR10, CIFAR100, and ImageNet-16-120 (ImageNet). CIFAR10 and CIFAR100 are two widely used datasets[3]. CIFAR-10 consists of 60,000 32x32 color images in 10 classes, with 6,000 images per class. There are 50,000 training images and 10,000 test images. CIFAR100 has 100 classes containing 600 images each. There are 500 training images and 100 testing images per class. ImageNet is provided by Chrabaszcz et al. (2017). NAS-Bench-201 chooses 4 nodes and 5 representative operation candidates for the operation set, which generates 15,625 cells/architectures as search space. In summary, NAS-Bench-201 enables researches to easily re-implement previous approaches via providing all architecture evaluation results. However, since these results are unavailable in real-world tasks, we take the evaluation time into account when computing the search time for a fair comparison.

### 6.2 BASELINES

We compare the proposed approach with the following baselines:

**Random search algorithms** It includes two baselines, random search (RS) (Bergstra & Bengio, 2012) and random search with parameter sharing (RSPS) (Li & Talwalkar, 2019).

**Reinforcement learning based algorithms** It includes one baseline, RL (Williams, 1992). RL is a method that introduces an architecture generator to generate well-performing architectures.

**Evolution based search algorithms** It includes one baseline: regularized evolution for architecture search (REA) (Real et al., 2019).

**Differentiable algorithms** It includes two baselines, DARTS (Liu et al., 2019) and GDAS (Dong & Yang, 2019). DARTS reformulates the search problem into a continues search problem. Following this work, GDAS updates a sub-graph rather than a whole graph for higher speeds.

**Hyper-parameter optimization algorithms** It includes one baseline, BOHB (Falkner et al., 2018). This approach combines the benefits of both Bayesian optimization and bandit-based methods.

We use the released code provided by Dong & Yang (2020) for baseline implementation. See Appendix A for more implementation and baseline details.

### 6.3 RESULTS

Table 1 shows the comparison between GT-NAS and the state-of-the-art approaches. The details of search time computing are described in Appendix A. RS is a naive baseline, which randomly selects architectures for full-training and evaluation. It is one of the most time-consuming approaches and achieves good accuracy improvements with 0.46 and 1.25 gains on CIFAR100 and ImageNet. RL adds learning into the search process and can explore more well-performing architectures. However, due to its unsteady training characteristic, RL does not beat RS on three datasets. REA contains two search stages: parent search and mutation. The second stage explores new architectures based on best-performing parent architectures. This approach achieves the highest performance with 93.72, 72.12, and 45.01 accuracy scores on CIFAR10, CIFAR100, and ImageNet. In conclusion, despite good results, these baselines require considerable time for architecture evaluation.

In comparison, there are also several approaches that achieve much better speed-up performance, such as RSPS, GDARTS and DARTS. However, the accuracy of architectures searched by these approaches

---

[3]https://www.cs.toronto.edu/ kriz/cifar.html

Table 1: Performance of different approaches on CIFAR10, CIFAR100, and ImageNet. "Time" means the search time. GT-NAS achieves competitive results with an extremely higher speed. In GT-NAS, all search steps can be finished within a few hours on a single GPU.

| Type | Model | CIFAR10 | | | CIFAR100 | | | ImageNet | | |
|------|-------|-----|---------|----------|-----|---------|----------|-----|---------|----------|
| | | Acc | Time(s) | Speed-up | Acc | Time(s) | Speed-up | Acc | Time(s) | Speed-up |
| w/o Search | ResNet | **93.97** | N/A | N/A | 70.86 | N/A | N/A | 43.63 | N/A | N/A |
| Search | RS | 93.63 | 216K | 1.0x | 71.28 | 460K | 1.0x | 44.88 | 1M | 1.0x |
| | RL | 92.83 | 216K | 1.0x | 70.71 | 460K | 1.0x | 44.10 | 1M | 1.0x |
| | REA | 93.72 | 216K | 1.0x | **72.12** | 460K | 1.0x | 45.01 | 1M | 1.0x |
| | BOHB | 93.49 | 216K | 1.0x | 70.84 | 460K | 1.0x | 44.33 | 1M | 1.0x |
| | RSPS | 91.67 | 10K | 21.6x | 57.99 | 46K | 21.6x | 36.87 | 104K | 9.6x |
| Gradient | GDAS | 93.36 | 22K | 12.0x | 67.60 | 39K | 11.7x | 37.97 | 130K | 7.7x |
| | DARTS | 88.32 | 23K | 9.4x | 67.34 | 80K | 5.8x | 33.04 | 110K | 9.1x |
| Kernel | GT-NAS | 93.42 | **7K** | **30.8x** | 71.42 | **18K** | **25.5x** | **45.35** | **31K** | **32.2x** |

is not high enough. Unlike these studies, GT-NAS achieves competitive results with an extremely higher speed. Compared to the time-consuming approaches, including RS, RL, REA, and BOHB, the proposed GT-NAS reduces search cost and brings 30.8x, 25.5x and 32.3x speed-up on CIFAR10, CIFAR100, and ImageNet. Compared to approaches with higher search speed, including RSPS, GDAS and DARTS, the proposed GT-NAS obtains large accuracy improvements. The searched architecture also outperforms ResNet with 0.56 and 1.72 accuracy improvements on CIFAR100 and ImageNet. Furthermore, all search steps only take a few hours on a single GPU in the proposed approach. For a better understanding, we visualize the relation between the kernel at initialization and test accuracy, as shown in Figure 4. The kernel is the average of a dot-product matrix in Eq. 1. Each dot represents a single architecture with its kernel (X-axis) and accuracy (Y-axis). We draw two plots with a different granularity of kernel ranges. On three datasets, higher kernels bring better average model performance. Since the kernel is decided by gradient values and correlations, this result also verifies our findings that both gradient values and gradient correlations matter.

**Universal to different architectures** NAS-Bench-201 contains over 100K architectures as search space, covering common network blocks. On such diverse dataset, the promising results of the proposed method show that GT-NAS is universal to different architectures.

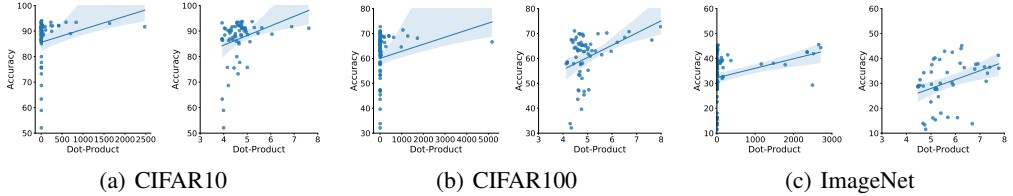

| (a) CIFAR10 | (b) CIFAR100 | (c) ImageNet |

Figure 4: Higher kernels brings more steady performance. Each dot represents a single architecture with its kernel (X-axis) and accuracy (Y-axis). For each dataset, we draw two plots with a different granularity of kernel ranges.

## 7 GENERALIZATION ON DIVERSE TASKS

We are also interested in whether GT-NAS can find better structures at acceptable cost on more datasets. To verify this, we conduct experiments on text classification tasks. Text classification includes two datasets, MRPC evaluating whether two sentences are semantically equivalent and RTE recognizing textual entailment. Two classification datasets are provided by Wang et al. (2019). See Appendix B for more dataset details.

Table 2: Results on text classification tasks. "Time" refers to search time.

| Models | MRPC | | RTE | |
|---|---|---|---|---|
| | Acc | Time(s) | Acc | Time(s) |
| Baseline | 92.08 | N/A | 83.51 | N/A |
| GT-NAS | **93.32 (+1.24)** | 0.4K | **83.75(+0.24)** | 2K |

## 7.1 SEARCH SPACE

For ease of implementation, we first generate some architectures as search space. We define several skip-connection structures, each with possible candidate architectures. We set the same layer number for all architectures: 12 encoder layers and 12 decoder layers. These architectures use the same hype-parameters as the baseline. Here are the details of different structures:

**Highway structure**   It connects all cells in a chain way. In each cell, there exists a highway connection from the first layer to the last layer. We adopt an average fusion way to merge all inputs to the last layer. The cell length ranges from 2 to 11 and there are 10 possible architectures.

**Look-ahead structure**   It connects all cells in a chain way. In each cell, the layer is connected in a chain way except for the last layer. The last layer takes all outputs of previous layers in a cell as inputs. We adopt an average fusion way to merge all inputs to the last layer. The cell length ranges from 2 to 11 and there are 10 possible architectures.

**DenseNet structure**   It splits the whole network into different cells (Huang et al., 2017b). The cell is connected in a fully connected way where the input of current cell comes from all outputs of previous cells. In each cell, the layer is connected in a chain way. The original network adopts a concatenation way to fuse the input vector together and elaborately designs a dimension shrinking and expanding policy. For better generalization, we replace the concatenation fusion with an average fusion way so that the dimension design efforts are not required. The cell length ranges from 2 to 11 and there are 10 possible architectures.

## 7.2 RESULTS

The results are shown in Table 2. The architectures searched by GT-NAS achieve better results on all datasets. Though pre-trained models are widely believed to be the state-of-the-art approaches, GT-NAS still obtains performance improvements over these strong baselines with $1.24$ and $0.24$ accuracy improvements on MRPC and RTE datasets. For two classification datasets, we choose top-2 and top-5 architectures for full training and evaluation. It proves that GT-NAS generalizes well on various datasets. In addition, these results show the importance of skip-connections in deep networks. Without adding any parameters, simply adjusting the way of connections can bring large performance improvements.

**Universal to different initialization approaches**   We conduct experiments on two different tasks, each with different initialization. The former uses Kaiming initialization, and the latter adopts Xaiver initialization. These two both are popular initialization methods. Table 1 and Table 2 demonstrate that our method is universal to different initialization approaches.

## 8 CONCLUSION

In this work, we propose a new explanation that gradient values and gradient correlations both matter in optimization. We believe that this can give a new insight into further optimization research. Following this new finding, we develop a fast yet effective approach which obtains competitive results with much higher speeds on a NAS benchmark dataset. Furthermore, GT-NAS generalizes well and brings improvements on diverse tasks including image classification and text classification.

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
