# OpenReview forum: "A Gradient-based Kernel Approach for Efficient Network Architecture Search"
_ICLR.cc/2021/Conference — Reject_

### Official Review · AnonReviewer1 · 2020-10-28
**This is a paper that maybe cannot reach the acceptance criteria.**

**Rating:** 4
**Confidence:** 4

**Review:**

This paper pointed out that both gradient correlations and gradient values have strong impacts on model training, which is an insightful finding. Based on this finding, they explore a simple yet effective network architecture search (NAS) approach. The new approach replaces the expensive “train-then-test” evaluation paradigm with a new lightweight function according to the gradient-based kernel at initialization. Experiments show the high speed of the approach.

Pros:
The finding that gradient correlations have strong impacts on model training is interesting to some extent

The idea that using gradient correlations to evaluate architectures is incremental, which breaks the limitation of the ‘train-then-test’ framework. As a result, it can help us take a step towards high-speed NAS.

Concerns:
The key concern about this paper is the lack of rigorous experimentation to study the usefulness of the proposed method. For a set of architectures, it is very important for a NAS method that the correlation between their ranking by the validation accuracy and their ranking by a NAS method. However, I did not see such an experiment in this paper. Further, comparison methods are not new enough. There is even no 2020’s method, even though there is a lot of excellent work in 2020 such as ISTA-NAS, PD-DARTS, which makes the experimental results not convincing enough.
The second concern is that the finding that gradient correlations have strong impacts on model training is similar to works such as SynFlow. I think it seems that the idea is not innovative enough. It would be better if the authors would compare these works thoroughly.

The third concern is about the figures and tables in the part of the experiment. For example, Table 1 reports that DARTS can achieve an accuracy of 88.32% on CIFAR10. However, our group also reconducted DARTS and achieve an accuracy of ~97%, which is similar to the results reported in many papers (e.g. DARTS, PC-DARTS). So I have a little doubt about the results in the table. For another example, figure 4 seems can only show the very limited correlation between kernel score and accuracy. Moreover, the two granularity here is hard to understand. I hope that you can explain it more specifically.

The organization of section 7 is a bit messy. For example, you list several search spaces in section 7.1, but you do not report how to use these search spaces. Are you use them separately or you aggregate them?

Minor comments:
Table2: It seems that you forget to report which is the “baseline”.
Eqution1: It would be more nice to see the definition of j.
Introduction:with largest kernels ->with the largest kernels
Section 4.2:a learning rate 1.0 -> a learning rate of 1.0
Section 6.1: enables researches to -> enables researchers to

---

### Official Review · AnonReviewer3 · 2020-10-30
**Unconvincing results**

**Rating:** 3
**Confidence:** 4

**Review:**

This paper defines a new scoring function for efficiently selecting between different network architectures, without requiring the expensive training of the models.  This scoring function is based on the expected dot-products between the stochastic gradient vectors corresponding to different pairs of training examples.  The paper shows that this method can much more quickly select between different architectures than previous network architecture search (NAS) algorithms, while attaining relatively similar performance.

Strengths
- Intuitively, it makes sense that higher correlations between the stochastic gradient updates will lead to less noisy updates, and thus hopefully result in a smoother optimization process and a better final model.

Weaknesses
- The main weaknesses of the paper in my opinion are lack of clarify and unconvincing empirical (and theoretical) results.
- The scoring function used for network selection is never clearly defined.  The text in section 5 simply says “average of the Gram matrix of gradient, i.e., P in Eq 1”, which is vague because P is a 3-dimensional tensor.   Furthermore, in the second paragraph of Section 5 it states that the gradients are normalized by “subtracting the average of sample gradients before calculation”, which is a very meaningful modification to the above-mentioned definition, which requires justification and explanation.  Lastly, it is claimed in the intro and throughout the paper that the metric unifies “gradient correlations” and “gradient values”, but this is never really explained.
- The proposed metric is not normalized in any way.  This suggests that larger models should automatically expect to have higher metric values.
- It seems the central argument of the paper is that there is a strong positive correlation between the proposed metric and the final test accuracy of the model.  However, the empirical validation of this result is unconvincing; the dots in Figure 4 seem very scattered/random, and the choice the zoom in at two different granularities for the x-axis in this plot is very mysterious—how were the x-axis ranges chosen?  It seems arbitrary.  And even with these hand-picked choices of x-axis, the positive correlation looks quite bad.  To make this convincing, it would be important to compute the actual correlations, and corresponding p-values, to see if these trends are significant.
- The take-aways from Figure 2 are quite unclear in my opinion.  Also, the figures are not clearly labeled.
- The claims that “GT-NAS is universal to different architectures” and that it is “universal to different initialization approaches” seem way too strong, given the limited empirical evidence.
- It was unclear to me how the math in Section 4.1 explains why the proposed metric is a good idea.  In order to better justify this metric choice, I would have hoped for a theorem formally analyzing in a simplified setting why this metric was a good idea.

Overall, due to the lack of clarity and unconvincing results, I recommend rejection for this paper.

---

### Official Review · AnonReviewer2 · 2020-11-03
**Interesting idea, but needs to be justified more**

**Rating:** 4
**Confidence:** 5

**Review:**

This paper consists of two parts, where it first studies the difficulty of neural network training, then proposes an efficient criterion for neural architecture search (NAS). In the first part, the authors propose two criteria, the average gradient absolute values and the average gradient correlations (across pairs of examples). It is claimed that both quantities are import for network training. In the second part, they propose to use gradient correlation as the reward for NAS, where it is assumed that the larger the gradient correlation the better the architecture is.

pros:

+The idea of using a kernel based metric as an efficient NAS criterion is interesting and have great potentials to simply expensive NAS procedures.

cons:

-The first part of the paper is intended to motivate the value of the gradient correlation, but it does not do a great job IMO. For one, the analysis of the gradient value part is not very interesting and seems obvious (thus redundant). It is also not super convincing to me that gradient correlation is either necessary nor sufficient for the model to achieve good test accuracy. More empirical results on this side will help, eg varying the dataset, base architecture and the diversity of model variants.

-Related to the previous point, is there a "theoretical" motivation for looking at gradient correlation?

-When studying the difficulty of training, it is extremely important to look at the training loss, rather than only showing the test accuracy. Fig 3 needs to include the training curves, otherwise it is hard to tell difficulty of training apart from generalization ability.

-The experimental results on NAS are not very convincing. All the numbers reported seem to be pretty weak in Table 2, including the baselines and the NAS variants. More specifically, on cifar10, none of the NAS variants outperform the resnet baseline. On imagenet, the models considered are too weak to be interesting (we know that a simple resnet18 model achieves close to 70% top1 acc). The results (both absolute and relative to baseline ones) need to be much improved.

In summary, I like the main idea of using the kernel statistics as a proxy for efficient NAS, but the paper needs to improve both the presentation and results to make it a significant contribution. Based on the current version, I vote for rejecting.

---

### Official Review · AnonReviewer5 · 2020-11-05
**[Borderline]**

**Rating:** 4
**Confidence:** 5

**Review:**

This paper shared some interesting observations about gradient values and correlations and utilized these findings in the context of neural architecture search to improve the search efficiency. To be specific, they use the gradients at initialization to filter out many "bad" architectures and then fully train the rest to find the best one.

Pros:
- The analysis of gradient correlations looks interesting to me, while I'm not sure whether it has been studied in previous works.
- The proposed approach is a kind training-free NAS method, which is an interesting direction in NAS.
- A comprehensive appendix.

Cons:
- The presentation should be improved. After reading the introduction, I'm still unclear about the proposed approach; if there could have a figure to explain some high-level concept of the proposed gradient-based kernel approach, it would be helpful to understand the paper.
- To make the paper self-contained, I would suggest the authors include the technical details in the main paper, such as how to sample?
- The proposed method is highly related to nas-without-training, where both of them can somehow estimate the quality of an architecture without the need of training. Therefore, I would encourage the authors compare the proposed method with https://github.com/BayesWatch/nas-without-training in a fair setting.
- The proposed method needs to sample-s-models and train-k-models. For a relatively small search space, it can be efficient. However, on a large search space such as 10^20 candidates, I feel it is much worse than differentiable methods. Coould the authors comment on that?
- A following question is the generalization ability of the proposed method on other search space. Would the authors try the proposed method on other search spaces, such as the size search space in NATS-Bench (an extension version of NAS-Bench-201) or NAS-Bench-101?

Some minor issues:
- In the second paragraph of Page 2, the reference for NAS-Bench-201 is incorrect, where the cited is NAS-Bench-101.
- In the "NAS" paragraph in the related work section, "Ying et al. (2019) propose a reinforcement learning based search policy" the reference seems incorrect, Ying et al. (2019) is NAS-Bench-101 instead of the RL-based NAS algorithm.
- In the paragraph, the authors said "the dependence on validation accuracy still leads to huge computation costs.", which is incorrect. The computation of validation accuracy is not expensive (depends on how to compute it). The huge computation costs are because that those approaches will need to fully train and evaluate each candidate architecture to get their validation accuracy. Also, the efficient search is due to the weight sharing approach, where the differentiable approach is a kind of popular direction.
- Therefore, I would suggest the authors revise the related works to fix these misleading parts and inaccurate references.

**Post Rebuttal**: The authors did not provide any responses, so I decreased the scores.

---

### Decision · Program_Chairs · 2021-01-07
**Final Decision**

**Decision:**

Reject

**Comment:**

This paper proposes a new criterion for neural architecture search that does not require the expensive step of training the model. The reviewers found the proposed approach of relying on gradient statistics promising. However, the reviewers found that the clarity of the paper needs to be improved and that the empirical evidence is too limited to support some of the claims.